# Bridging Visual Affective Gap: Borrowing Textual Knowledge by Learning from Noisy Image-Text Pairs

## ABSTRACT

Visual emotion recognition (VER) is a longstanding field that has garnered increasing attention with the advancement of deep neural networks. Although recent studies have achieved notable improvements by leveraging the knowledge embedded within pre-trained visual models, the lack of direct association between factual-level features and emotional categories, called the "affective gap", limits the applicability of pre-training knowledge for VER tasks. On the contrary, the explicit emotional expression and high information density in textual modality eliminate the "affective gap". Therefore, we propose borrowing the knowledge from the pre-trained textual model to enhance the emotional perception of pre-trained visual models. We focus on the factual and emotional connections between images and texts in noisy social media data, and propose **P**artitioned **A**daptive **C**ontrastive **L**earning (**PACL**) to leverage these connections. Specifically, we manage to separate different types of samples and devise distinct contrastive learning strategies for each type. By dynamically constructing negative and positive pairs, we fully exploit the potential of noisy samples. Through comprehensive experiments, we demonstrate that bridging the "affective gap" significantly improves the performance of various pre-trained visual models in downstream emotion-related tasks.

## CCS CONCEPTS

• **Information systems** → **Sentiment analysis**; • **Computing methodologies** → **Computer vision tasks**.

## KEYWORDS

visual emotion recognition, affective gap, factual connection, emotional connection, contrastive learning

**ACM Reference Format:**

Anonymous Authors. 2024. Bridging Visual Affective Gap: Borrowing Textual Knowledge by Learning from Noisy Image-Text Pairs. In *Proceedings of the 32nd ACM International Conference on Multimedia (MM'24), October 28-November 1, 2024, Melbourne, Australia.* ACM, New York, NY, USA, 10 pages. https://doi.org/10.1145/nnnnnnn.nnnnnnn

## 1 INTRODUCTION

Visual emotion recognition (VER) aims at identifying human emotions towards different visual stimuli [41]. As a vital facet of human engagement with the world, perceiving emotions through visual

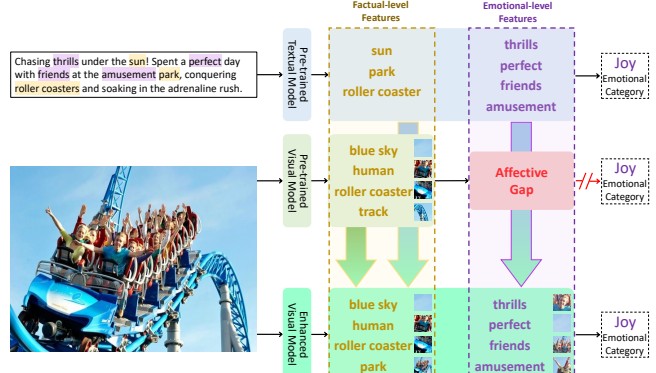

**Figure 1: Illustration of the "affective gap" in the pre-trained visual model and how our proposed method bridges it using knowledge from the pre-trained textual model.**

cues is progressively emerging as a pivotal challenge on the path toward the next generation of artificial general intelligence [22, 50]. Therefore, it has drawn increasing attention in recent years [9, 24] and exhibited broad applications in opinion mining [37], business intelligence [14], and autonomous driving [17].

Prior studies in VER have greatly enhanced their performance by leveraging the powerful generalizable knowledge of pre-trained models [61]. However, a misalignment exists between the objectives of their pre-training and downstream tasks, leading to the "affective gap" phenomenon, as depicted in Fig. 1. Specifically, the commonly adopted pre-training tasks, such as classification on ImageNet [4] and CLIP [28]-type vision-language pretraining, encourage models to acquire non-trivial factual-level features under semantic guidance. Yet, these features lack a direct association with the emotional categories, leading to the incomplete applicability of pre-training knowledge for downstream VER tasks. To bridge the "affective gap", the pre-trained model needs the capability of encoding emotional-level features, which serves as an intermediate link between factual-level features and emotion categories.

An intuitive way to acquire such capability is learning from large-scale emotional annotated datasets, while it would encounter two primary challenges in practice. Firstly, the perception subjectivity of emotion [25, 63] results in high annotation costs, restricting the scale of high-quality datasets. Secondly, the sparse information density of visual data [3, 39] causes a dispersed distribution of emotional information, requiring models to extract features across multiple regions and scales, thereby increasing the demand for computational resources. However, language has natural advantages in these two aspects compared with vision, as it explicitly conveys emotions through specific words and possesses higher and more consistent information density [34]. These advantages allow pre-trained textual models to simultaneously learn generalizable factual-level and emotional-level features from large corpora in

unsupervised manners, eliminating the "affective gap" in textual modality.

Inspired by this discovery, we propose to bridge the visual "affective gap" by enhancing pre-trained visual models with the unified factual-level and emotional-level features from pre-trained textual models, as illustrated in Fig. 1. To transfer this knowledge from texts to images, we focus on image-text datasets where samples generally possess relatively strong factual and emotional connections between their images and texts. Specifically, we adopt TumEmo [45], an image-text dataset sourced from social media. Due to the inherent factual inconsistencies and emotional discrepancies of user-generated posts [46], the dataset inevitably incorporates certain noisy samples that are factual-mismatched, emotional-mismatched samples, or both. Given such characteristics, we propose **P**artitioned **A**daptive **C**ontrastive **L**earning (**PACL**) to leverage these connections within all samples. It comprises three stages. Firstly, we quantitatively assess the factual and emotional connections between the image and text of each sample, dividing the dataset into separate partitions. It allows us to formulate distinct learning strategies for different types of data. Secondly, we cluster on two unimodal subsets of the dataset, revealing the inter-sample relationships from both factual and emotional aspects. Thirdly, we design an adaptive contrastive strategy by utilizing the outcomes from the preceding stages. We dynamically filter out false negative pairs and reconstruct false positive pairs for different dataset partitions, thereby ensuring the consistent superiority of positive pairs over negative pairs, both factually and emotionally.

We follow the network architecture of CLIP [28], with its textual encoder initialized by SKEP [36] and its visual encoder initialized by the to-be-enhanced visual model. In this manner, the enhanced visual model can encode emotional-level features from an image while retaining the original perception of factual-level features. It effectively bridges the "affective gap" in visual models and significantly improves their performance in downstream VER tasks.

The overall contributions of this paper are three-fold:

- We harness the advantages of language over vision. As far as we know, this is the first attempt to bridge the "affective gap" in pre-trained visual models with pre-trained textual knowledge.
- We propose a method called PACL. It leverages the factual and emotional connections within noisy image-text pairs by dynamically constructing contrastive pairs for different types of samples.
- We conduct extensive experiments on **six** VER benchmarks and **four** kinds of downstream tasks with **four** pre-trained visual models. The results demonstrate the consistent improvements achieved by our method, underscoring the effectiveness and necessity of bridging the "affective gap".

## 2 RELATED WORKS

### 2.1 Visual Emotion Recognition

Vision plays a crucial role in human emotional perception, and research on VER has been conducted for over two decades [64]. Early studies mainly focus on designing hand-crafted features based on psychology and art theories, including low-level elements [48], mid-level principals [54, 60], and high-level adjective-noun pairs [1]. Later on, the rapid advancements in deep learning reveal the superiority of learning-based features [29, 53], catalyzing the emergence

of numerous methods built upon it [43, 52]. As their performance gradually approaches the bottleneck, researchers shift focus towards perceiving emotions at multiple regions and scales[44, 51]. More recently, various correlative subfields witness increased attention, such as personalized prediction [47, 63], domain adaptation [59, 65] and zero-shot learning [49, 55].

The rapid development of VER in recent years is inseparable from the emergence of pre-trained models. The non-trivial knowledge embedded in these models has significantly aided them in perceiving emotions [58]. However, the misalignment between pre-training objectives and downstream VER tasks leads to the "affective gap"- the absence of intermediate emotional-level features between factual-level features and emotional categories. To tackle this problem, Feng *et al.* [10] imitate human's visual sentiment perception mechanism and design six pre-training tasks to guide the model in learning factual-level and emotional-level features from scratch. Although achieving impressive performance gains for several backbones, they expand the original network size by threefold and demand extensive annotated data from diverse datasets, imposing significant computational costs on the training process. In contrast to them, we enhance the emotional perception of pre-trained visual models on top of the original factual-level knowledge, significantly reducing the demand for data.

### 2.2 Learning from Text Supervision

Language and vision have their respective pros and cons in different aspects. In recent years, many studies have explored harnessing the advantages of language to enhance visual models. Out of them, CLIP [28] demonstrates the efficacy of text supervision in guiding the acquisition of robust and transferable visual representations. Following works are proposed by considering such as fine-grained alignment [3, 30, 66], additional supervision [7, 66], external knowledge [11, 33] and non-contrastive objective [67].

Inspired by their success, EmotionCLIP [58] transfers this paradigm to VER, by guiding the visual model in learning emotion-level features with a subject-aware context encoding and a sentiment-guided contrastive learning. It incorporates the knowledge of a pre-trained textual model during training, similar to ours, but only employs it for computing relationships between samples. Additionally, their approach requires substantial video and text data strictly correlated in both factual and emotional aspects. However, they haven't released their training data yet, and to our knowledge, no publicly available dataset meets this requirement. Consequently, subsequent works following them entail not only significant training costs, akin to [10], but also necessitate additional expenses for data collection. In comparison, our comprehensive utilization of pre-training textual knowledge and the noisy image-text pairs mitigates the demands on both the quantity and quality of training data and enables stronger performance on downstream tasks.

## 3 METHOD

The pipeline of our proposed PACL is demonstrated in Fig. 2. It comprises three stages: Dataset Partition (Section 3.1), Unimodal Sample Cluster (Section 3.2) and Adaptive Contrastive Learning (Section 3.3). We provide a detailed explanation of each stage below.

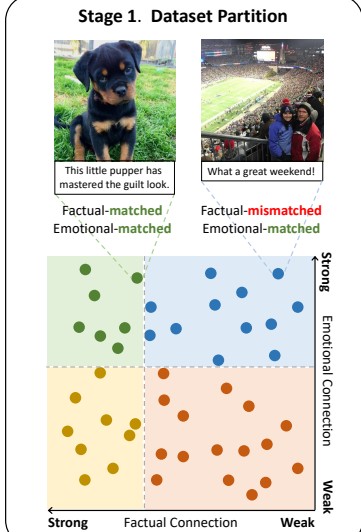
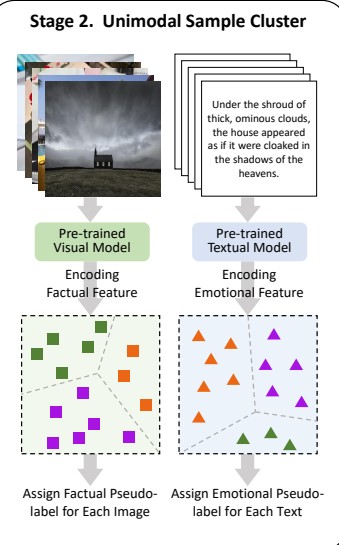
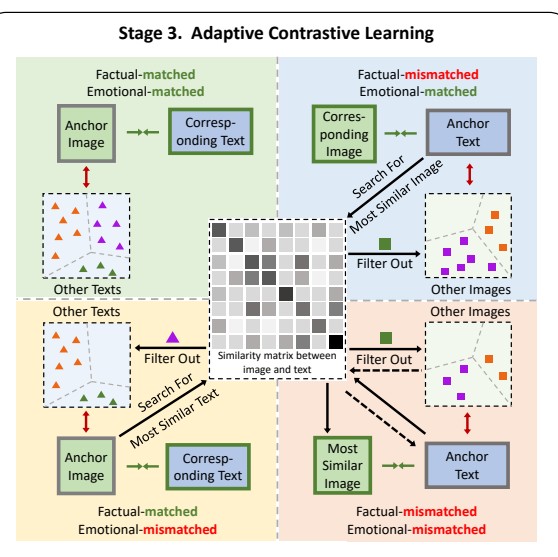

**Figure 2: Overview of PACL's three-stage pipeline. In the first stage, we divide the dataset into four partitions according to whether samples are factual-matched, emotional-matched, or not. In the second stage, we perform k-means clustering on image and text separately to obtain the factual and emotional pseudo-label for each of them. In the third stage, we guide the knowledge transfer from text to image with an adaptive contrastive learning. It dynamically constructs positive and negative pairs for different partitions by leveraging the pseudo-labels of samples.**

## 3.1 Dataset Partition

We represent a total of $N$ image-text pairs in the training dataset as $\{(v_i, t_i)|i = 1, 2, \cdots, N\}$. Due to being collected from social media, images and texts from the same sample may suffer from inherent factual inconsistencies and emotional discrepancies. To prevent the model from learning misleading factual or emotional connections, we first divide the dataset into four partitions by quantitatively assessing the factual and emotional connections of each sample.

We employ CLIP [28] as the grounding evaluator of the factual connection, with its visual and textual encoder denoted by $C_v(\cdot)$ and $C_t(\cdot)$. Given a sample $(v_i, t_i)$, we consider it is factual-matched if its CLIP-similarity score is larger than a threshold $\sigma$:

$$\frac{C_v(v_i) \odot C_t(t_i)}{||C_v(v_i)|| \cdot ||C_t(t_i)||} > \sigma. \tag{1}$$

Otherwise, we consider it factual-mismatched.

We employ DeepSentiBank [2], denoted by $D(\cdot)$ and pre-trained BERTweet [26], denoted by $B(\cdot)$, as the grounding evaluator of the emotional connection. For sample $(v_i, t_i)$, we first transform its image $v_i$ into the top-3 adjective-noun pairs $D(v_i)$. Then, we determine whether it is emotional-matched based on a calculation similar to the above:

$$\frac{B(D(v_i)) \odot B(t_i)}{||B(D(v_i))|| \cdot ||B(t_i)||} > \sigma. \tag{2}$$

Following this, we divide the dataset into four partitions: **1. strong-coupled samples** that are factual-matched and emotional-matched; **2. partial-coupled samples** that are emotional-matched but factual-mismatched; **3. partial-coupled samples** that are factual-matched but emotional-mismatched; **4. weak-coupled samples** that are factual-mismatched and emotional-mismatched. We

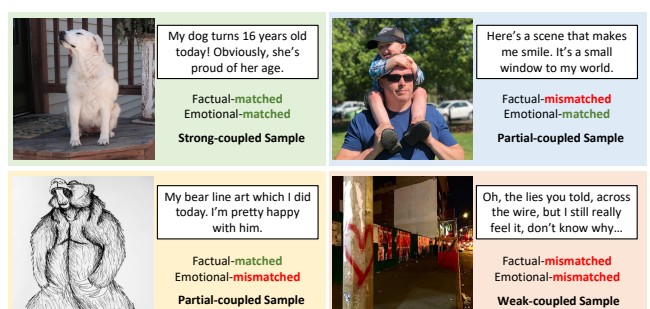

**Figure 3: Samples from four partitions of TumEmo [45].**

display a sample from each partition in Fig. 3 for intuitive understanding. It should be noted that we employ CLIP, DeepSentiBank, and BERTweet to simulate the partition conducted by humans, which can be replaced with other off-the-shelf tools.

## 3.2 Unimodal Sample Cluster

To optimize the construction of sample pairs in the subsequent contrastive learning, we utilize the knowledge embedded in the pre-trained visual and textual models to explore the unimodal inter-sample relationships from both factual and emotional perspectives. We adopt the network architecture of CLIP, as shown in Fig. 4. In this stage, none of the gradients are involved in the calculation.

For the visual encoder, we initialize it with parameters from a pre-trained visual model, denoted by $V_\theta(\cdot)$, which serves as the enhanced target of PACL. Leveraging its factual-level knowledge, we encode factual features for all images, $\{v_i|i = 1, 2, \cdots, N\}$, and conduct k-means clustering inside its feature space. It yields $K$

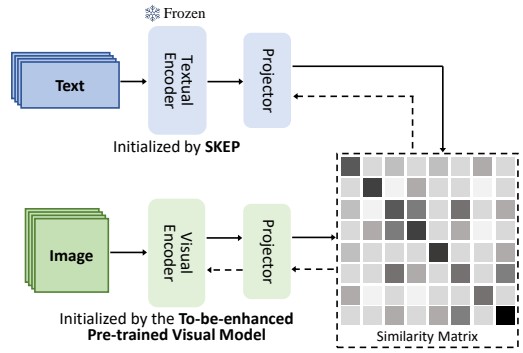

**Figure 4: Network architecture of PACL.**

clusters, where $K$ is a hyperparameter describing the cluster granularity, and each image is assigned a cluster label. We denote the label of image $v_i$ as its factual pseudo-label $f_i$. Thus, we obtain the factual relationship between images $v_i, v_j$: they are factually similar if $f_i = f_j$, and dissimilar if $f_i \neq f_j$.

For the textual encoder, we initialize it with parameters from SKEP[36], denoted by $T_\xi(\cdot)$. SKEP is an unsupervised sentiment knowledge-enhanced pre-trained model focusing more on emotional features. Leveraging this characteristic, we encode emotional features for all texts, $\{t_i|i = 1, 2, \cdots, N\}$, and perform k-means clustering that also yields $K$ clusters. Similarly, we denote the cluster label of text $t_i$ as its emotional pseudo-label $e_i$.

### 3.3 Adaptive Contrastive Learning

With the outcomes of the two preparation stages, we devise an adaptive contrastive learning that constructs positive and negative pairs under different strategies for each dataset partition. In this stage, we treat SKEP as the knowledge source and keep the parameters of the textual encoder frozen. We update the parameters of the visual encoder through contrastive learning to guide it in learning the generalizable factual-level and emotional-level knowledge from textual modality.

*3.3.1* **Preliminary**. We employ contrastive learning between images and texts multiple times. Denoting the features of $(v_i, t_i)$ as $z_i^v$ and $z_i^t$, here we provide a general form of contrastive loss:

$$
\mathcal{L} = -\frac{1}{2} \log \frac{\sum_{j \in p^+(v_i)} exp((z_i^v \odot z_j^t)/\tau)}{\sum_{j \in \{p^+(v_i) \cup p^-(v_i)\}} exp((z_i^v \odot z_j^t)/\tau)}
$$
$$
-\frac{1}{2} \log \frac{\sum_{j \in p^+(t_i)} exp((z_j^v \odot z_i^t)/\tau)}{\sum_{j \in \{p^+(t_i) \cup p^-(t_i)\}} exp((z_j^v \odot z_i^t)/\tau)}. \quad (3)
$$

$p^+(\cdot), p^-(\cdot)$ represent the positive and negative samples of the anchor. $\tau$ is the temperature hyperparameter. This loss comprises two components, with either image $v_i$ or text $t_i$ serving as the anchor. Taking the example of image $v_i$, the objective of Eq. (3) is to bring it closer to $p^+(v_i)$ while pushing it away from $p^-(v_i)$.

*3.3.2* **Strong-coupled Samples**. We first conduct contrastive learning solely on this partition to facilitate the alignment between the feature spaces of visual encoder $V_\theta(\cdot)$ and textual encoder $T_\xi(\cdot)$. Since samples are both factual-matched and emotional-matched, we treat the images and texts from the same samples as positive

pairs, and those from different samples as negative pairs. It enables the model to assign higher cosine similarity scores to image-text pairs with strong factual and emotional connections. Denoting the projector of the visual branch as $q_v(\cdot)$ and it of the textual branch as $q_t(\cdot)$, the features of image $v_i$ and text $t_i$ are computed as:

$$
z_i^v = q_v(V_\theta(v_i)), \quad z_i^t = q_t(T_\xi(t_i)). \quad (4)
$$

By substituting Eq. (4) into Eq. (3), we obtain the contrastive loss of this dataset partition:

$$
\mathcal{L}_1 = \mathcal{L}(p^+(v_i) = p^+(t_i) = \{i\},
$$
$$
p^-(v_i) = p^-(t_i) = \{j|j \neq i\}). \quad (5)
$$

*3.3.3* **Partial-coupled Samples**. These samples are from two dataset partitions. We discuss them together due to their common properties, that their images and texts have strong connections only in one of the factual and emotional aspects. To leverage these connections, we construct positive pairs in the same way as strong-coupled samples. However, since partial-coupled samples no longer guarantee the advantages of positive pairs over negative pairs in both factual and emotional aspects, the former construction of negative pairs is sub-optimal.

For the partition where samples are emotional-matched but factual-mismatched, we take $(v_i, t_i)$ as an example. Compared to $v_i$, the image $v_j$ from another sample may demonstrate a stronger factual connection with $t_i$, and in this case, treating $(v_i, t_i)$ as a positive pair while $(v_j, t_i)$ as a negative pair compromises the factual-level learning of the visual model. Therefore, we devise a mapping $m_f(\cdot)$ that searches for images that might possess strong factual connections with text $t_i$ to avoid inappropriate negative pairs:

$$
m_f(t_i) = \{v_j|l = \underset{k}{\arg\max}\, z_k^v \odot z_i^t, f_j = f_l\}. \quad (6)
$$

It first finds the most matching image, denoted by $v_l$, based on the cosine similarity scores and then selects images with factual pseudo-label same as $f_l$. We also define an inversion of $m_f(\cdot)$ that searches for texts based on image $v_i$:

$$
m_f^{-1}(v_i) = \{t_j|v_i \in m_f(t_j)\}. \quad (7)
$$

Subsequently, the negative samples of $v_i$ are all other texts with the exclusion of $m_f^{-1}(v_i)$, and the negative samples of $t_i$ are all other images with the exclusion of $m_f(t_i)$. Thereby, the contrastive loss of this partition is:

$$
\mathcal{L}_2 = \mathcal{L}(p^+(v_i) = p^+(t_i) = \{i\},
$$
$$
p^-(v_i) = \{j|j \neq i, t_j \notin m_f^{-1}(v_i)\}, \quad (8)
$$
$$
p^-(t_i) = \{j|j \neq i, v_j \notin m_f(t_i)\}).
$$

For the partition where samples are factual-matched but emotion-mismatched, we devise a symmetric $m_e(\cdot)$ that searches for texts that possess strong emotional connections with image $v_i$:

$$
m_e(v_i) = \{t_j|l = \underset{k}{\arg\max}\, z_i^v \odot z_k^t, e_j = e_l\}, \quad (9)
$$

and also an inversion of $m_e(\cdot)$ that searches for images based on text $t_i$:

$$
m_e^{-1}(t_i) = \{v_j|t_i \in m_e(v_j)\} \quad (10)
$$

Similarly, the negative samples of $v_i$ ($t_i$) are all other texts (images) with the exclusion of $m_e(v_i)$ ($m_e^{-1}(t_i)$). The contrastive loss of this partition is:

$$\mathcal{L}_3 = \mathcal{L}(p^+(v_i) = p^-(t_i) = \{i\},$$
$$p^-(v_i) = \{j | j \neq i, t_j \notin m_e(v_i)\}, \quad (11)$$
$$p^-(t_i) = \{j | j \neq i, v_j \notin m_e^{-1}(t_i)\}).$$

*3.3.4 Weak-coupled Samples.* Since samples of this partition are factual-mismatched and emotional-mismatched, constructing positive pairs as before would disrupt the semantic relationships learned by the model [19] and lead to degenerations in downstream performance. Therefore, we reconstruct positive pairs by leveraging the learned correlation between images and texts in previous partitions. Taking sample ($v_i, t_i$) as an example, we select positive samples according to the cosine similarity scores:

$$p^+(v_i) = \{\underset{k}{\arg\max}\ z_i^v \odot z_k^t\}, \quad (12)$$

$$p^+(t_i) = \{\underset{k}{\arg\max}\ z_k^v \odot z_i^t\}. \quad (13)$$

For negative pairs, we follow the strategies adopted for partial-coupled samples. We simultaneously apply $m_f(\cdot)$ and $m_e(\cdot)$ to filter out potential false negative pairs with either strong factual or emotional connections. As a result, the contrastive loss is:

$$\mathcal{L}_4 = \mathcal{L}(p^+(v_i) = \{\underset{k}{\arg\max}\ z_i^v \odot z_k^t\},$$
$$p^+(t_i) = \{\underset{k}{\arg\max}\ z_k^v \odot z_i^t\},$$
$$p^-(v_i) = \{j | j \neq i, t_j \notin m_f^{-1}(v_i) \cup m_e(v_i)\}, \quad (14)$$
$$p^-(t_i) = \{j | j \neq i, v_j \notin m_f(t_i) \cup m_e^{-1}(t_i)\}).$$

*3.3.5 Total Loss.* We devise a progressive learning schedule to guide the model to learn from strong-coupled samples to weak-coupled samples step by step. In the first one-third of epochs, we only incorporate the strong-coupled samples into training:

$$\mathcal{L}_{total} = \mathcal{L}_1. \quad (15)$$

In the second one-third of epochs, we incorporate both the strong-coupled and partial-coupled samples into training:

$$\mathcal{L}_{total} = \mathcal{L}_1 + \mathcal{L}_2 + \mathcal{L}_3. \quad (16)$$

In the last one-third of epochs, we incorporate all samples into training:

$$\mathcal{L}_{total} = \mathcal{L}_1 + \mathcal{L}_2 + \mathcal{L}_3 + \mathcal{L}_4. \quad (17)$$

## 4 EXPERIMENT

### 4.1 Datasets

We conduct training of PACL on **TumEmo** [45]. It contains 200k image-text pairs crawled from social media. Its samples are annotated based on their emotions. Since we emphasize leveraging the factual and emotional connections between images and texts, we avoid using knowledge from the annotations.

In the downstream testing, we adopt six publicly available VER datasets that are annotated according to diverse emotion theories. **FI** [53] is labeled according to Mikels' psychological study [23]. **Emotion-6** and **UnbiasedEmo** [24] contain 6 common emotion

categories. **WebEmo** [24] is labeled according to Parrot's hierarchical emotion model [27], which contains 2 basic categories at the first level, 7 categories at the second level and 25 fine-grained categories at the third level. **Emotic** [16] comprises 26 fine-grained emotion categories, with each image assigned multiple emotion labels, and 10-scale VAD (Valence-Arousal-Dominance) [32] ratings. **IAPS** [18] is labeled by 9-scale VAD ratings.

### 4.2 Implementation Details

All of our implementations are based on Pytorch. We employ three visual models pre-trained on ImageNet [4], including ResNet-50 [13], ViT-base [8], Swin-base [21], and one visual model pre-trained by CLIP [28] (ViT-clip) as the target models for enhancement. Most of our experiments adopt SKEP [36] as the pre-trained textual model. In Section 5, we also evaluate the enhancement effect of other textual models [6, 20, 31]. In the first stage of PACL, we set the partition threshold $\sigma$ to 0.7. In the second stage, we set cluster granularity $K$ to 2. In the third stage, we set the contrastive temperature $\tau$ to 0.07, the initial learning rate of ResNet-50 to 1e-3, ViT-base and ViT-clip to 1e-5, Swin-base to 2e-5, projection layers $p_v(\cdot)$, $p_t(\cdot)$ to 1e-3. We train the model for 30 epochs with a minibatch size of 64. We adopt AdamW optimizer with a cosine schedule to decay the learning rate. In the downstream testing, we adopt the original partition for training, validation, and testing if provided. Otherwise, we randomly divide the dataset in a ratio of 8:1:1. All the experiments are conducted on four NVIDIA 4090 GPUs.

### 4.3 Compared Baselines

Two other works also focus on enhancing pre-trained visual models in perceiving emotions. **Feng et al.** [10] factorize human perception into three steps and devise a pre-training method to imitate it. Their pre-training tasks include colorization, super resolution, scene recognition, jigsaw puzzles, ANP prediction, and image captioning, requiring 10.5M images with annotations. We reproduce it under all of our settings for comparison. **EmotionCLIP** [58] adopts vision-language pre-training to extract emotional representations from verbal and nonverbal communication. It leverages the pre-trained knowledge from CLIP [28] and DistilBert [31]. It is trained on 1M video-text pairs with the annotation of sentiment score and human bounding boxes. Since the training data is not publicly available yet, we compare PACL with it under pre-trained ViT-clip, which is the only model released. **PACL** is our proposed model, it leverages the pre-trained knowledge from SKEP [36] and is trained on 200K image-text pairs without any additional annotations. PACL has **the loosest requirements for training data** among them.

### 4.4 Enhancement Comparison

We adopt four kinds of VER downstream tasks. **1**. In **single-label learning**, models are required to classify each image into a single emotional category. **2**. In **hierarchical-label learning**, models are required to predict categories for three emotion levels simultaneously. We report accuracy (Acc) and weighted-F1 (F1) for the first two tasks. **3**. In **multi-label learning**, models are required to predict multiple emotion categories for each image. We report the mean of average precision (mAP) and area under roc curve (AUC) of categories. **4**. In **VAD learning**, models are required to predict

**Table 1: Emotional enhancement of visual models on single-label learning datasets including FI [53], Emotion-6 [24], UnbiasedEmo [24], hierarchical-label learning dataset WebEmo [24], multi-label learning dataset Emotic [16], and VAD learning datasets including Emotic [16] and IAPS [18]. Accuracy (Acc↑), weighted-F1 (F1↑), mean of average precision (mAP↑), area under roc curve (AUC↑), mean of mean square error (MSE↓), mean of R square ($R^2$ ↑) are reported for these datasets. Acc and F1 are followed by the number of total categories, and MSE is multiplied by $10^2$. The best results are marked in bold.**

| Model | Single-label | | | | | | Hierarchical-label | | | | | | Multi-label | | VAD | | | |
| | FI | | Emotion-6 | | UnbiasedEmo | | WebEmo | | | | | | Emotic | | Emotic | | IAPS | |
| | Acc-8 | F1-8 | Acc-6 | F1-6 | Acc-6 | F1-6 | Acc-2 | F1-2 | Acc-6 | F1-6 | Acc-25 | F1-25 | mAP | AUC | MSE | $R^2$ | MSE | $R^2$ |
|---|---|---|---|---|---|---|---|---|---|---|---|---|---|---|---|---|---|---|
| *Linear Evaluation:* | | | | | | | | | | | | | | | | | | |
| ResNet-50 | 57.72 | 56.72 | 46.47 | 45.18 | 60.20 | 60.08 | 66.25 | 66.21 | 41.08 | 39.03 | 24.94 | 22.80 | 25.85 | 66.36 | 2.825 | 0.1267 | 2.130 | 0.2790 |
| ResNet-50 + Feng *et al.* | 58.39 | 57.88 | 47.54 | 46.48 | 62.17 | 61.27 | 65.82 | 65.71 | 40.63 | 38.34 | 24.57 | 22.14 | **27.32** | 67.49 | 2.867 | 0.1279 | 2.253 | 0.2685 |
| ResNet-50 + PACL | **59.50** | **59.03** | **51.14** | **50.59** | **66.78** | **66.39** | **67.53** | **67.42** | **43.60** | **40.95** | **25.85** | **24.32** | 27.02 | **67.60** | **2.646** | **0.1404** | **1.982** | **0.3104** |
| ViT-base | 62.08 | 62.06 | 54.85 | 54.24 | 74.67 | 74.24 | 70.72 | 70.67 | 45.90 | 43.67 | 27.69 | 26.10 | 30.16 | 68.26 | 2.763 | 0.1265 | 2.010 | 0.2936 |
| ViT-base + Feng *et al.* | 64.12 | 64.06 | 57.02 | 56.73 | 77.71 | 77.54 | 70.55 | 70.51 | 45.34 | 45.09 | 27.37 | 27.14 | 33.01 | 69.30 | 2.733 | 0.1293 | 1.991 | 0.3088 |
| ViT-base + PACL | **64.77** | **64.64** | **58.25** | **58.03** | **78.95** | **78.91** | **71.98** | **71.88** | **48.32** | **46.01** | **28.64** | **27.89** | **33.40** | **70.68** | **2.435** | **0.1466** | **1.903** | **0.3319** |
| Swin-base | 63.63 | 63.50 | 53.29 | 51.27 | 74.30 | 73.98 | 70.39 | 70.23 | 46.39 | 44.25 | 28.82 | 27.41 | 31.27 | 68.09 | 2.715 | 0.1308 | 1.966 | 0.2872 |
| Swin-base + Feng *et al.* | 65.03 | 64.74 | 55.47 | 52.84 | 76.30 | 75.67 | 70.12 | 70.03 | 46.37 | 44.36 | 28.56 | 27.17 | 32.00 | 69.08 | 2.665 | 0.1335 | 1.909 | 0.2948 |
| Swin-base + PACL | **65.70** | **65.58** | **57.13** | **56.79** | **79.93** | **79.67** | **72.14** | **72.13** | **47.96** | **45.81** | **29.89** | **27.95** | **32.91** | **70.19** | **2.451** | **0.1449** | **1.794** | **0.3285** |
| ViT-clip | 71.74 | 71.61 | 64.43 | 64.15 | 85.53 | 85.24 | 78.41 | 77.85 | 54.86 | 53.68 | 37.36 | 36.12 | 38.50 | 72.67 | 2.313 | 0.1681 | 1.560 | 0.3473 |
| ViT-clip + Feng *et al.* | 64.12 | 64.06 | 57.02 | 56.73 | 77.71 | 77.54 | 70.55 | 70.51 | 45.34 | 45.09 | 27.37 | 27.14 | 33.01 | 69.30 | 2.733 | 0.1293 | 1.991 | 0.3088 |
| ViT-clip + EmotionCLIP | 72.63 | 72.32 | 64.72 | 64.57 | 84.36 | 84.06 | 78.77 | 78.22 | 55.28 | 54.35 | 37.12 | 35.82 | 34.63 | 72.33 | 2.303 | 0.1711 | 1.538 | 0.3478 |
| ViT-clip + PACL | **73.17** | **72.53** | **66.08** | **66.01** | **86.72** | **85.97** | **79.51** | **79.33** | **56.13** | **55.59** | **37.40** | **37.03** | **39.05** | **72.97** | **2.249** | **0.1744** | **1.506** | **0.3535** |
| *Fine-tuning:* | | | | | | | | | | | | | | | | | | |
| ResNet-50 | 65.14 | 64.84 | 53.46 | 53.25 | 73.68 | 73.15 | 73.81 | 73.80 | 48.88 | 48.34 | 30.85 | 29.89 | 28.36 | 68.43 | 2.757 | 0.1337 | 2.114 | 0.2842 |
| ResNet-50 + Feng *et al.* | 66.69 | 66.55 | 58.21 | 58.05 | **77.47** | **76.90** | 74.43 | 74.38 | 49.66 | 49.39 | 32.63 | 31.01 | 28.44 | 68.20 | 2.646 | 0.1408 | 2.082 | 0.2954 |
| ResNet-50 + PACL | **67.11** | **66.79** | **58.60** | **58.16** | 77.30 | 76.65 | **74.76** | **74.75** | **50.18** | **49.21** | **32.72** | **31.42** | **30.39** | **70.12** | **2.531** | **0.1460** | **1.927** | **0.3130** |
| ViT-base | 69.57 | 69.56 | 57.45 | 57.09 | 77.63 | 77.51 | 76.25 | 76.08 | 52.49 | 51.01 | 33.88 | 32.52 | 32.52 | 71.73 | 2.633 | 0.1374 | 1.975 | 0.3016 |
| ViT-base + Feng *et al.* | 69.73 | 69.60 | 60.15 | 59.38 | 81.03 | 81.00 | 76.34 | 76.11 | 53.39 | 52.41 | 33.98 | 33.62 | 33.12 | 71.87 | 2.603 | 0.1451 | 1.964 | 0.3129 |
| ViT-base + PACL | **70.91** | **70.68** | **60.92** | **60.65** | **82.57** | **82.26** | **77.61** | **77.74** | **54.87** | **54.50** | **34.75** | **34.61** | **35.09** | **72.24** | **2.378** | **0.1511** | **1.860** | **0.3325** |
| Swin-base | 70.24 | 70.33 | 55.57 | 55.27 | 78.29 | 78.02 | 75.83 | 75.75 | 51.57 | 51.05 | 34.19 | 33.10 | 33.11 | 71.61 | 2.607 | 0.1369 | 1.922 | 0.2961 |
| Swin-base + Feng *et al.* | 72.04 | 71.68 | 58.62 | 58.41 | 82.29 | 82.17 | 76.71 | 76.53 | 53.90 | 53.62 | 34.50 | 34.08 | 32.97 | 71.53 | 2.580 | 0.1461 | 1.861 | 0.3115 |
| Swin-base + PACL | **73.21** | **73.17** | **60.04** | **59.72** | **83.55** | **83.34** | **77.19** | **77.06** | **54.69** | **54.32** | **35.33** | **35.27** | **36.11** | **72.83** | **2.407** | **0.1530** | **1.779** | **0.3344** |
| ViT-clip | 77.33 | 77.25 | 64.89 | 64.60 | 88.16 | 88.04 | 79.32 | 79.25 | 55.47 | 54.99 | 39.71 | 38.82 | 38.78 | 72.90 | 2.246 | 0.1729 | 1.523 | 0.3562 |
| ViT-clip + Feng *et al.* | 69.73 | 69.60 | 60.15 | 59.38 | 81.03 | 81.00 | 76.34 | 76.11 | 53.39 | 52.41 | 33.98 | 33.62 | 33.12 | 71.87 | 2.603 | 0.1451 | 1.964 | 0.3129 |
| ViT-clip + EmotionCLIP | 77.62 | 77.40 | 65.09 | 64.77 | 89.44 | 88.81 | 80.20 | 80.16 | 56.14 | 55.83 | 39.67 | 38.70 | 36.54 | 72.52 | 2.264 | 0.1719 | 1.476 | 0.3575 |
| ViT-clip + PACL | **80.55** | **80.28** | **66.91** | **66.65** | **90.14** | **90.13** | **81.84** | **81.57** | **56.35** | **56.02** | **39.99** | **39.21** | **39.36** | **73.03** | **2.213** | **0.1784** | **1.425** | **0.3664** |

the continuous values in the 'Valence', 'Arousal', and 'Dominance' dimensions. We report the mean of mean squared error (MSE) and R squared ($R^2$) of the three dimensions. Besides, we conduct experiments under two evaluation protocols: linear evaluation and fine-tuning. In both protocols, we set the learning rate of the linear classification head to 1e-2 and train for 40 epochs. For parameters of the backbone network, we keep them frozen under linear evaluation and set the initial learning rate of ResNet-50 to 1e-4, ViT-base, Swin-base, and ViT-clip to 1e-5 under fine-tuning. The other settings of downstream testing are aligned with Section 4.2.

The experimental results are presented in Table 1. We can observe that PACL consistently improves all the pre-trained visual models on all tasks by a large margin. In contrast, the effectiveness of Feng *et al.* is limited under linear evaluation, especially in hierarchical-label learning. It highlights the advantages of harnessing textual knowledge in perceiving emotions at multiple levels. Additionally, since they start the pre-training from scratch, they can not leverage the generalizable factual knowledge in the pre-trained visual models. For instance, PACL achieves significant performance gains by starting from ViT-clip instead of ViT-base, underscoring the effectiveness and scalability of bridging the "affective gap". EmotionCLIP, on the other hand, also undergoes poor performance on some tasks, specifically, multi-label learning. We attribute this to

the misalignment between its pre-training and testing. Although it explicitly models the dependencies between context and humans, it may be misled by the massive communications between humans in training data and learns emotional representations heavily dependent on humans. However, the multi-label learning dataset Emotic emphasizes the emotion perception from both context and humans, resulting in the undesirable performance of EmotionCLIP. On the contrary, PACL leverages diverse image-text pairs and the unified factual and emotional features embedded in texts, naturally encouraging visual models to perceive emotions in different regions and scales, thereby promoting universal enhancements on all tasks.

## 4.5 Comparison with SOTA

To further demonstrate the necessity and effectiveness of bridging the "affective gap", we present a brief view of the current SOTA methods in two downstream tasks.

*4.5.1 **Classification on FI, WebEmo**.* Following previous methods [40], we group eight categories of FI into two coarse-grained parent categories. We report the classification accuracy of SOTA methods on FI and WebEmo at multiple emotion levels in Table 2. It should be noted that the models are tested on each level separately, different from the settings in hierarchical-label learning in 4.4. As shown in Table 2, ViT-clip enhanced by PACL outperforms other

**Table 2: Accuracy of SOTA methods in classification at different emotion levels on FI [53] and WebEmo [24]. The best results are marked in bold, and the second-best results are underlined.**

| Model | FI | | WebEmo | | |
|---|---|---|---|---|---|
| | Acc-2 | Acc-8 | Acc-2 | Acc-6 | Acc-25 |
| Sentibank [1] | 56.47 | 44.49 | - | - | - |
| DeepSentibank [2] | 64.39 | 53.16 | - | - | - |
| PCNN [52] | 75.34 | 56.16 | - | - | - |
| Zhu *et al.* [68] | 84.26 | 73.03 | - | - | - |
| Rao *et al.* [43] | 87.51 | 75.46 | - | - | - |
| WSCNet [42] | 86.74 | 70.07 | 79.43 | 52.61 | 32.75 |
| PDANet [62] | 87.25 | 72.13 | 80.96 | 53.46 | 32.82 |
| Zhang and Xu [56] | 90.97 | 75.91 | 82.47 | 53.88 | 33.01 |
| MDAN [40] | 91.08 | 76.41 | 82.72 | 55.65 | 34.28 |
| SimEmotion [5] | 95.42 | 80.33 | - | - | - |
| ViT-clip + PACL | 95.27 | 80.55 | 84.06 | 57.42 | 40.31 |
| ViT-clip + PACL + PDANet | **96.06** | **80.83** | **84.74** | **58.11** | **40.50** |

methods in most cases. Additionally, since our enhanced model does not incorporate additional parameters besides the backbone network, it can be combined with other methods for further improvements. By adding a polarity-consistent regression loss [62], our enhanced model achieves the best performances across all scenarios. These results validate the efficacy and scalability of PACL.

**Table 3: Accuracy of SOTA methods in zero-shot classification on FI [53] and Emotion-6 [24]. [C] denotes the names of emotion categories.**

| Model | Text Prompt | FI | Emotion-6 |
|---|---|---|---|
| | | Acc-8 | Acc-6 |
| Zhang *et al.* [57] | — | 21.73 | 22.74 |
| Kodirov *et al.* [15] | — | 19.35 | 26.53 |
| Sung *et al.* [35] | — | 18.49 | 28.10 |
| Zhan *et al.* [55] | — | 24.51 | 30.49 |
| Ye *et al.* [49] | — | **36.03** | **33.57** |
| CLIP [28] | A photo with [C] emotion. | 20.71 | 10.16 |
| ResNet-50 + PACL | A photo with [C] emotion. | 25.58 | 29.46 |
| CLIP [28] | The photo makes me feel so [C]! | 17.45 | 13.78 |
| ResNet-50 + PACL | The photo makes me feel so [C]! | 31.72 | 30.23 |

*4.5.2* ***Zero-shot on FI, Emotion-6***. Since we adopt the network architecture of PACL the same as CLIP [28], our enhanced models are naturally capable of conducting zero-shot emotion classification by leveraging the learned emotional connections between images and texts. We report the zero-shot classification accuracy of the enhanced ResNet-50 and the SOTA methods in Table 3. Similar to CLIP, our enhanced model is also sensitive to text prompts. Therefore, we test its performances under both regular and manually designed prompts. From the results, we observe that the enhanced ResNet-50 significantly outperforms CLIP by learning from much fewer image-text pairs. We attribute this to the difference between the types of training data. PACL learns from social media posts, which contain generally strong emotional connections between images and texts, as well as more frequent occurrences of textual emotional expressions. Compared with other methods, the enhanced

**Table 4: Ablation experiments on dataset partition and sample usage. For factual (emotional) partition, we conduct contrastive learning by treating factual-matched (emotional-matched) samples as strong-coupled and factual-mismatched (emotional-mismatched) samples as partial-coupled. "Both" denotes factual + emotional.**

| Partition | Sample Usage | UnbiasedEmo | |
|---|---|---|---|
| | | Acc-6 | F1-6 |
| Initial weight | | 60.20 | 60.08 |
| None | All | 60.53 | 60.01 |
| Factual | Factual-matched | 63.49 | 62.91 |
| Factual | Factual-matched + Factual-mismatched | 65.13 | 64.71 |
| Emotional | Emotional-matched | 64.47 | 63.77 |
| Emotional | Emotional-matched + Emotional-mismatched | 65.46 | 64.83 |
| Both | Strong-coupled | 64.14 | 64.10 |
| Both | Strong-coupled + Partial-coupled | 65.79 | 65.38 |
| Both | Strong-coupled + Partial-coupled + Weak-coupled | **66.78** | **66.39** |

ResNet-50 achieves the second-best performance on both datasets. Although zero-shot emotion classification is not our primary goal, the satisfactory performance achieved by the enhanced ResNet-50 provides us with the intuitive advantages of learning from social media posts.

## 5 ANALYSIS

In this section, we validate the effectiveness of each component of PACL. All the experiments are conducted by adopting pre-trained ResNet-50 as the backbone under linear evaluation.

### 5.1 Ablation Study

We conduct ablation experiments in Table 4 to explore the influences of different dataset partitions and sample usages. Firstly, we do not apply partition strategies and conduct contrastive learning by treating all the samples as strong-coupled. It results in minor enhancements for the pre-trained model since TumEmo contains certain noisy samples with either weak factual or emotional connections. In these cases, the vanilla contrastive learning is sub-optimal. Next, we apply partition based only on factual connections, treating factual-matched samples as strong-coupled and factual-mismatched samples as partial-coupled. PACL enhances the pre-trained model by a large margin with solely factual-matched samples and further improves its performance by utilizing factual-mismatched samples. We also apply partition based only on emotional connections with the same settings, which brings a similar trend. By dynamically constructing contrastive pairs, we succeed in harnessing these noisy samples. Notably, although the downstream task is emotion-related, applying partition based on either emotional or factual connections can both lead to significant enhancements, demonstrating the cruciality of proper handling of both factual-mismatched and emotional-mismatched samples. Finally, by simultaneously applying factual and emotional partition as PACL, the pre-trained model progressively learns from factual and emotional connections of different kinds of data, achieving the best result. The above results indicate the effectiveness of PACL in fully exploiting the dataset.

**Table 5: Experiments of hyperparameter selection. Each experiment for a specific hyperparameter is carried out independently, with other non-tested hyperparameters set to their adopted values. $\sigma, K, \tau$ are from Eq. (1), Section 3.2, Eq. (3), respectively.**

| Partition Threshold $\sigma$ | FI | | Emotion-6 | | WebEmo | |
|---|---|---|---|---|---|---|
| | Acc-8 | F1-8 | Acc-6 | F1-6 | Acc-2 | F1-2 |
| 0 | 57.08 | 56.74 | 46.59 | 45.11 | 65.51 | 65.21 |
| 0.3 | 58.50 | 58.03 | 49.10 | 47.74 | 66.36 | 66.33 |
| 0.5 | 58.58 | 57.96 | 49.46 | 48.00 | 67.02 | 66.94 |
| 0.7 (**adopted**) | **59.59** | **59.03** | **51.14** | **50.59** | **67.53** | **67.42** |
| 0.9 | 56.64 | 56.06 | 45.75 | 44.31 | 65.08 | 64.94 |
| 1 | 39.56 | 37.77 | 36.17 | 33.72 | 58.54 | 58.11 |

| Cluster Granularity $K$ | FI | | Emotion-6 | | WebEmo | |
|---|---|---|---|---|---|---|
| | Acc-8 | F1-8 | Acc-6 | F1-6 | Acc-2 | F1-2 |
| 2 (**adopted**) | **59.59** | **59.03** | 51.14 | 50.59 | **67.53** | **67.42** |
| 3 | 59.23 | 58.51 | 51.02 | 50.34 | 67.30 | 67.12 |
| 6 | 58.46 | 57.66 | **51.98** | **51.78** | 67.44 | 67.35 |
| 10 | 57.98 | 57.82 | 49.34 | 49.08 | 66.82 | 66.50 |
| 15 | 57.33 | 56.67 | 47.90 | 47.13 | 66.60 | 66.38 |
| 25 | 56.08 | 55.82 | 45.63 | 43.56 | 65.21 | 64.79 |

| Contrastive Temperature $\tau$ | FI | | Emotion-6 | | WebEmo | |
|---|---|---|---|---|---|---|
| | Acc-8 | F1-8 | Acc-6 | F1-6 | Acc-2 | F1-2 |
| 0.01 | 59.43 | 59.12 | 49.34 | 48.89 | 66.53 | 66.44 |
| 0.03 | **59.74** | **59.38** | 50.19 | 49.83 | 67.22 | 67.10 |
| 0.05 | 59.45 | 58.77 | 50.96 | **50.62** | 67.03 | 66.79 |
| 0.07 (**adopted**) | 59.59 | 59.03 | **51.14** | 50.59 | **67.53** | **67.42** |
| 0.5 | 58.81 | 58.53 | 50.27 | 49.80 | 66.91 | 66.73 |
| 1 | 58.45 | 58.13 | 49.61 | 49.29 | 66.80 | 66.55 |

## 5.2 Hyperparameter Selections

To probe the influence of hyperparameters, we conduct the experiments in Table 5.

*5.2.1* **Partition Threshold** $\sigma$. It controls the precision and size of each partition. By adopting a small threshold, the strong-coupled samples can not guarantee the generally strong factual and emotional connections between images and texts. It disrupts the alignment between the visual and textual feature spaces [19]. On the other hand, a large threshold limits the number of strong-coupled samples, which is also not conducive to the alignment. Consequently, we adopt a medium threshold that provides sufficient high-quality strong-coupled samples. Specifically, 22% samples are strong-coupled, 51% samples are partial-coupled, and 27% samples are weak-coupled under $\sigma = 0.7$.

*5.2.2* **Cluster Granularity** $K$. It balances the distances between intra-cluster and inter-cluster samples, thereby influencing the reconstruction of negative pairs for partial-coupled and weak-coupled samples. Fine-grained clustering guarantees the correctness of filtered negative pairs, while coarse-grained clustering emphasizes the correctness of the remaining. According to the experimental results, we adopt a small $K$. This indicates that the advantages of filtering out additional true negative pairs outweigh the potential drawbacks of ignoring false negative pairs. Interestingly, the enhanced model achieves the best results at $K = 6$ on Emotion-6. We attribute this to the alignment in the granularities of clustering and downstream classification. However, we still adopt $K = 2$ due to its general advantages across all datasets.

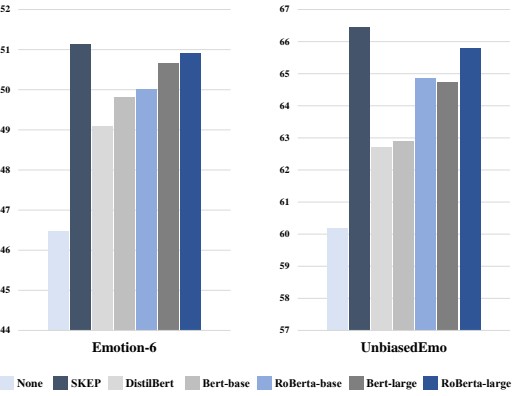

**Figure 5: Classification accuracy of the enhanced visual model on Emotion-6 [24] and UnbiasedEmo [24] under text supervisions from different pre-trained textual models. Specifically, we employ SKEP [36], DistilBert [31], Bert-base [6], RoBerta-base [20], Bert-large [6] and ReBerta-large [20].**

*5.2.3* **Contrastive Temperature** $\tau$. It controls the strength of penalties on negative samples [38]. As shown in the results, PACL is not sensitive to its selection. The enhanced model achieves slightly superior results under smaller temperatures compared to larger selections. Therefore, we adopt $\tau = 0.07$ according to results and related works [12, 28].

## 5.3 Different Text Supervisions

We evaluate the generalizability of PACL by employing the knowledge from various pre-trained textual models to enhance the visual model. As shown in Fig. 5, different pre-trained textual models bring consistent performance improvements on both Emotion-6 and UnbiasedEmo. It validates the universal advantages of textual models in encoding emotional-level features compared to visual models. Additionally, we can observe a positive correlation between the representative capability of the textual model and its enhancement effectiveness through PACL on the visual model. It demonstrates the scalability of PACL relative to the textual models. With the evolution of large language models in recent years, the potential of PACL can be further exploited.

## 6 CONCLUSION

In this paper, we focus on bridging the visual "affective gap" of the pre-trained visual models. It arises from the misalignment between the objectives of pre-training and downstream tasks. Inspired by the advantages of language, we propose **P**artitioned **A**daptive **C**ontrastive **L**earning (**PACL**) that enhances visual models by leveraging the unified factual-level and emotional-level features from the pre-trained textual model. PACL learns from the factual and emotional connections from noisy image-text pairs collected from social media by dataset partition, unimodal sample cluster, and adaptive contrastive learning. Through extensive experiments, we demonstrate the effectiveness and scalability of PACL in enhancing the emotional perception of visual models, validating the importance of bridging the "affective gap".

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
