# OpenReview forum: "Bridging Visual Affective Gap: Borrowing Textual Knowledge by Learning from Noisy Image-Text Pairs"
_acmmm.org/ACMMM/2024/Conference — MM2024 Oral_

### Official Review · Reviewer_bCZC · 2024-05-17

**Rating:** 4
**Confidence:** 2

**Summary:**

This paper proposes a Partitioned Adaptive Contrastive Learning (PACL) method to leverage the factual and emotional connections between images and texts in noisy social media data. By dynamically constructing negative and positive pairs, the model fully exploits the potential of noisy samples. Experiments on six VER benchmarks and four kinds of downstream tasks with four pre-trained visual models demonstrate the effectiveness of the proposed method.

**Strengths:**

1. The studied topic, i.e., bridging the “affective gap” in pre-trained visual models with pre-trained textual knowledge, is interesting.
2. The experiments are convincing. The ablation studies on different components are well conducted to demonstrate the effectiveness of the proposed method.

**Limitations:**

1. Figures 2 and 3 require a clear indication of the dotted and solid lines.
2. More explanation is needed on how the combination of "ViT-clip + PACL + PDANet" in Table 2 implements and contributes to performance improvements.
3. Why the proposed ResNet-50 + PACL underperforms against Ye et al. [49] in zero-shot classification, as shown in Table 3?

**Suitability:**

3

---

### Official Review · Reviewer_HaRo · 2024-05-22

**Rating:** 3
**Confidence:** 3

**Summary:**

This paper identifies the affective gap as the lack of direct association between factual-level features and emotional categories in visual models. The authors propose a contrastive learning-based method that uses noisy image-text pairs to create dynamic positive and negative pairs for contrastive learning. This approach separates different types of samples and applies distinct strategies to each, fully exploiting the potential of noisy data. Experiments are conducted following [10].

**Strengths:**

1.	The idea of introducing image-text data is interesting and will benefit the community.
2.	Extensive experiments demonstrate the effectiveness of proposed method.
3.	Ablation studies verify that each component is an optimal solution.

**Limitations:**

1.	Affective gap is a long-standing problem in visual emotion recognition, and any progress in this area would be highly valuable. The authors have some efforts to introduce textual knowledge to narrow this gap. However, the paper lacks a clear and precise definition of the "affective gap," making it challenging to assess the extent to which it has been bridged. It would significantly strengthen the paper to provide both quantitative and qualitative metrics (rather than a single sample of image-text pair) that can be used to evaluate the gap.
2.	The results of the basic backbone are somewhat unusual. If the reviewer correctly understands the experimental setting, the accuracy of the fine-tuned ResNet50 on the FI dataset (65.14%) is lower than those reported in other works (67.00% from [10] and 67.53% from [A, B]). This discrepancy suggests that the proposed PACL method offers only a limited performance improvement (+0.11%). Additionally, the performance of [10] reported in this work is inconsistent with the results from the original paper. Could the authors provide more detailed information about their implementation? Clarifying these details would help in understanding the experimental conditions and ensuring the reproducibility of the results. This additional information is crucial for accurately assessing the effectiveness of the proposed method.
3.	The paper lacks some critical experimental details, and no code has been provided, which raises concerns about the reproducibility of the results.

[A] EmoSet: A Large-scale Visual Emotion Dataset with Rich Attributes, ICCV23

[B] Stimuli-Aware Visual Emotion Analysis, TIP21

Although I appreciate the interesting idea of this work, but there are issues about some of inconsistent performance that can not align with previous works and some bold statement that is not verified by evidence. This make the reviewer leans to reject.

**Suitability:**

3

---

### Official Review · Reviewer_Ziow · 2024-05-24

**Rating:** 4
**Confidence:** 4

**Summary:**

This paper presents Partitioned Adaptive Comparison Learning (PACL), which bridges the visual “affective gap” of pre-trained visual models by augmenting them with unified factual-level and emotional-level features from the pre-trained textual model. PACL learns from the factual and emotional connections from noisy image-text pairs collected from social media by dataset partition, unimodal sample cluster, and adaptive contrastive learning.

**Strengths:**

1.	This paper is well-written and easy to follow.
2.	Experiments are also solid, providing extensive comparisons as well as ablation studies.

**Limitations:**

1.	Regarding the proposed progressive learning methodology that divides the entire training process equally into three parts, is there a basis for such a division? Has a more optimal training approach been explored?
2.	The clustering method in Stage 2 uses consistent clustering granularity for factual and emotional, and is puzzled as to whether the granularity must be aligned between the two. It is recommended to add more comparative experiments to verify the authors' settings in the article
3.	It would be helpful if the papers could elaborate on limitations suggesting future directions and potential avenues for future research.
4.	While the paper is generally well-written, it would greatly benefit from a careful proofreading to address the presence of some typographical errors in the manuscript, e.g. the lack of a period after Equation (10).

**Suitability:**

3

---

### Official Review · Reviewer_KMgJ · 2024-06-02

**Rating:** 6
**Confidence:** 4

**Summary:**

proposed Partitioned Adaptive Contrastive Learning (PACL) for Visual emotion recognition.

**Strengths:**

managed to separate different types of samples and devised distinct contrastive learning strategies for each type.
By dynamically constructing negative and positive pairs, exploited the potential of noisy samples.

**Limitations:**

1. Analyzing computational complexity.
2. Provide reproduction code.
3. Where there are correlations between emotions, visualize the distribution of emotions and validate the model's ability to discriminate emotions.

**Suitability:**

3

---

### Meta-Review · Area_Chair_M979 · 2024-06-29

**Recommendation:** Accept (Oral)
**Confidence:** 3

**Metareview:**

We would like to thank the authors for answering the questions by the reviewers in the rebuttal. Reviewers have considered your statements and most reviewers felt that their questions/concerns have been addressed properly. As such, we are happy to accept the paper.